# Learning on Large-scale Text-attributed Graphs via Variational Inference

**Jianan Zhao**[1,3*], **Meng Qu**[1,3*], **Chaozhuo Li**[2†], **Hao Yan**[4], **Qian Liu**[5], **Rui Li**[6], **Xing Xie**[2], **Jian Tang**[1,7,8†]
[1]Mila - Québec AI Institute, [2]Microsoft Research Asia, [3]Université de Montréal
[4]Central South University, [5]Sea AI Lab, [6]Dalian University of Technology
[7]HEC Montréal, [8]Canadian Institute for Advanced Research (CIFAR)

## Abstract

This paper studies learning on text-attributed graphs (TAGs), where each node is associated with a text description. An ideal solution for such a problem would be integrating both the text and graph structure information with large language models and graph neural networks (GNNs). However, the problem becomes very challenging when graphs are large due to the high computational complexity brought by training large language models and GNNs together. In this paper, we propose an efficient and effective solution to learning on large text-attributed graphs by fusing graph structure and language learning with a variational Expectation-Maximization (EM) framework, called GLEM. Instead of simultaneously training large language models and GNNs on big graphs, GLEM proposes to alternatively update the two modules in the E-step and M-step. Such a procedure allows training the two modules separately while simultaneously allowing the two modules to interact and mutually enhance each other. Extensive experiments on multiple data sets demonstrate the efficiency and effectiveness of the proposed approach [1].

## 1 Introduction

Graphs are ubiquitous in the real world. In many graphs, nodes are often associated with text attributes, resulting in text-attributed graphs (TAGs) (Yang et al., 2021). For example, in social graphs, each user might have a text description; in paper citation graphs, each paper is associated with its textual content. Learning on TAG has become an important research topic in multiple areas including graph learning, information retrieval, and natural language processing.

In this paper, we focus on a fundamental problem, learning effective node representations, which could be used for a variety of applications such as node classification and link prediction. Intuitively, a TAG is rich in textual and structural information, both of which could be beneficial for learning good node representations. The textual information presents rich semantics to characterize the property of each node, and one could use a pre-trained language model (LM) (e.g., BERT (Devlin et al., 2019)) as a text encoder. Meanwhile, the structural information preserves the proximity between nodes, and connected nodes are more likely to have similar representations. Such structural relationships could be effectively modeled by a graph neural network (GNN) via the message-passing mechanism. In summary, LMs leverage the local textual information of individual nodes, while GNNs use the global structural relationship among nodes.

An ideal approach for learning effective node representations is therefore to combine both the textual information and graph structure. One straightforward solution is to cascade an LM-based text encoder and GNN-based message-passing module and train both modules together. However, this method suffers from severe scalability issues. This is because the memory complexity is proportional to the graph size as neighborhood texts are also encoded. Therefore, on real-world TAGs where nodes are densely connected, the memory cost of this method would become unaffordable.

To address such a problem, multiple solutions have been proposed. These methods reduce either the capacity of LMs or the size of graph structures for GNNs. More specifically, some studies choose to

---

*The first two authors contributed equally. †Corresponding authors.
[1]Codes are available at `https://github.com/AndyJZhao/GLEM`.

fix the parameters of LMs without fine-tuning them (Liu et al., 2020b). Some other studies reduce graph structures via edge sampling and perform message-passing only on the sampled edges (Zhu et al., 2021; Li et al., 2021a; Yang et al., 2021). Despite the improved scalability, reducing the LM capacity or graph size sacrifices the model effectiveness, leading to degraded performance of learning effective node representation. Therefore, we are wondering whether there exists a scalable and effective approach to integrating large LMs and GNNs on large text-attributed graphs.

In this paper, we propose such an approach, named **G**raph and **L**anguage Learning by **E**xpectation **M**aximization GLEM. In the GLEM, instead of simultaneously training both the LMs and GNNs, we leverage a variational EM framework (Neal & Hinton, 1998) to alternatively update the two modules. Take the node classification task as an example. The LM uses local textual information of each node to learn a good representation for label prediction, which thus models label distributions conditioned on text attributes. By contrast, for a node, the GNN leverages the labels and textual encodings of surrounding nodes for label prediction, and it essentially defines a global conditional label distribution. The two components are optimized to maximize a variational lower bound of the log-likelihood function, which can be achieved by alternating between an E-step and an M-step, where at each step we fix one component to update the other one. This separate training framework significantly improves the efficiency of GLEM, allowing it to scale up to real-world TAGs. In each step, one component presents pseudo-labels of nodes for the other component to mimic. By doing this, GLEM can effectively distill the local textual information and global structural information into both components, and thus GLEM enjoys better effectiveness in node classification. We conduct extensive experiments on three benchmark datasets to demonstrate the superior performance of GLEM. Notably, by leveraging the merits of both graph learning and language learning, GLEM-LM achieves on par or even better performance than existing GNN models, GLEM-GNN achieves new state-of-the-art results on ogbn-arxiv, ogbn-product, and ogbn-papers100M.

## 2 RELATED WORK

Representation learning on text-attributed graphs (TAGs) (Yang et al., 2021) has been attracting growing attention in graph machine learning, and one of the most important problems is node classification. The problem can be directly formalized as a text representation learning task, where the goal is to use the text feature of each node for learning. Early works resort to convolutional neural networks (Kim, 2014; Shen et al., 2014) or recurrent neural networks Tai et al. (2015). Recently, with the superior performance of transformers (Vaswani et al., 2017) and pre-trained language models (Devlin et al., 2019; Yang et al., 2019), LMs have become the go-to model for encoding contextual semantics in sentences for text representation learning. At the same time, the problem can also be viewed as a graph learning task that has been vastly developed by graph neural networks (GNNs) (Kipf & Welling, 2017; Velickovic et al., 2018; Xu et al., 2019; Zhang & Chen, 2018; Teru et al., 2020). A GNN takes numerical features as input and learns node representations by transforming representations and aggregating them according to the graph structure. With the capability of considering both node attributes and graph structures, GNNs have shown great performance in various applications, including node classification and link prediction. Nevertheless, LMs and GNNs only focus on parts of observed information (i.e., textual or structural) for representation learning, and the results remain to be improved.

There are some recent efforts focusing on the combination of GNNs and LMs, which allows one to enjoy the merits of both models. One widely adopted way is to encode the texts of nodes with a fixed LM, and further treat the LM embeddings as features to train a GNN for message passing. Recently, a few methods propose to utilize domain-adaptive pretraining (Gururangan et al., 2020) on TAGs and predict the graph structure using LMs (Chien et al., 2022; Yasunaga et al., 2022) to provide better LM embeddings for GNN. Despite better results, these LM embeddings still remain unlearnable in the GNN training phase. Such a separated training paradigm ensures the model scalability as the number of trainable parameters in GNNs is relatively small. However, its performance is hindered by the task and topology-irrelevant semantic modeling process. To overcome these limitations, endeavors have been made (Zhu et al., 2021; Li et al., 2021a; Yang et al., 2021; Bi et al., 2021; Pang et al., 2022) to co-train GNNs and LM under a joint learning framework. However, such co-training approaches suffer from severe scalability issues as all the neighbors need to be encoded by language models from scratch, incurring huge extra computation costs. In practice, these models restrict the message-passing to very few, e.g. 3 (Li et al., 2021a; Zhu et al., 2021), sampled first-hop neighbors, resulting in severe information loss.

To summarize, existing methods of fusing LMs and GNNs suffer from either unsatisfactory results or poor scalability. In contrast to these methods, GLEM uses a pseudo-likelihood variational framework to integrate an LM and a GNN, which allows the two components to be trained separately, leading to good scalability. Also, GLEM encourages the collaboration of both components, so that it is able to use both textual semantics and structural semantics for representation learning, and thus enjoys better effectiveness.

Besides, there are also some efforts using GNNs for text classification. Different from our approach, which uses GNNs to model the observed structural relationship between nodes, these methods assume graph structures are unobserved. They apply GNNs on synthetic graphs generated by words in a text (Huang et al., 2019; Hu et al., 2019; Zhang et al., 2020) or co-occurrence patterns between texts and words (Huang et al., 2019; Liu et al., 2020a). As structures are observed in the problem of node classification in text-attributed graphs, these methods cannot well address the studied problem.

Lastly, our work is related to GMNN Qu et al. (2019), which also uses a pseudo-likelihood variational framework for node representation learning. However, GMNN aims to combine two GNNs for general graphs and it does not consider modeling textual features. Different from GMNN, GLEM focuses on TAGs, which are more challenging to deal with. Also, GLEM fuses a GNN and an LM, which can better leverage both structural features and textual features in a TAG, and thus achieves state-of-the-art results on a few benchmarks.

## 3 BACKGROUND

In this paper, we focus on learning representations for nodes in TAGs, where we take node classification as an example for illustration. Before diving into the details of our proposed GLEM, we start with presenting a few basic concepts, including the definition of TAGs and how LMs and GNNs can be used for node classification in TAGs.

### 3.1 TEXT-ATTRIBUTED GRAPH

Formally, a TAG $\mathcal{G}_S = (V, A, \mathbf{s}_V)$ is composed of nodes $V$ and their adjacency matrix $A \in \mathbb{R}^{|V| \times |V|}$, where each node $n \in V$ is associated with a sequential text feature (sentence) $\mathbf{s}_n$. In this paper, we study the problem of node classification on TAGs. Given a few labeled nodes $\mathbf{y}_L$ of $L \subset V$, the goal is to predict the labels $\mathbf{y}_U$ for the remaining unlabeled objects $U = V \setminus L$.

Intuitively, node labels can be predicted by using either the textual information or the structural information, and representative methods are language models (LMs) and graph neural networks (GNNs) respectively. Next, we introduce the high-level ideas of both methods.

### 3.2 LANGUAGE MODELS FOR NODE CLASSIFICATION

Language models aim to use the sentence $\mathbf{s}_n$ of each node $n$ for label prediction, resulting in a text classification task (Socher et al., 2013; Williams et al., 2018). The workflow of LMs can be characterized as below:

$$p_\theta(\mathbf{y}_n | \mathbf{s}_n) = \text{Cat}(\mathbf{y}_n \mid \text{softmax}(\text{MLP}_{\theta_2}(\mathbf{h}_n))); \qquad \mathbf{h}_n = \text{SeqEnc}_{\theta_1}(\mathbf{s}_n), \qquad (1)$$

where $\text{SeqEnc}_{\theta_1}$ is a text encoder such as a transformer-based model (Vaswani et al., 2017; Yang et al., 2019), which projects the sentence $\mathbf{s}_n$ into a vector representation $\mathbf{h}_n$. Afterwards, the node label distribution $\mathbf{y}_n$ can be simply predicted by applying an $\text{MLP}_{\theta_2}$ with a softmax function to $\mathbf{h}_n$.

Leveraging deep architectures and pre-training on large-scale corpora, LMs achieve impressive results on many text classification tasks (Devlin et al., 2019; Liu et al., 2019). Nevertheless, the memory cost is often high due to large model sizes. Also, for each node, LMs solely use its own sentence for classification, and the interactions between nodes are ignored, leading to sub-optimal results especially on nodes with insufficient text features.

### 3.3 GRAPH NEURAL NETWORKS FOR NODE CLASSIFICATION

Graph neural networks approach node classification by using the structural interactions between nodes. Specifically, GNNs leverage a message-passing mechanism, which can be described as:

$$p_\phi(\mathbf{y}_n | A) = \text{Cat}(\mathbf{y}_n \mid \text{softmax}(\mathbf{h}_n^{(L)})); \qquad \mathbf{h}_n^{(l)} = \sigma(\text{AGG}_\phi(\text{MSG}_\phi(\mathbf{h}_{\text{NB}(n)}^{(l-1)}), A)), \qquad (2)$$

where $\phi$ denotes the parameters of GNN, $\sigma$ is an activation function, $\mathrm{MSG}_\phi(\cdot)$ and $\mathrm{AGG}_\phi(\cdot)$ stand for the message and aggregation functions respectively, $\mathrm{NB}(n)$ denotes the neighbor nodes of $n$. Given the initial text encodings $\mathbf{h}_n^{(0)}$, e.g. pre-trained LM embeddings, a GNN iteratively updates them by applying the message function and the aggregation function, so that the learned node representations and predictions can well capture the structural interactions between nodes.

With the message-passing mechanism, GNNs are able to effectively leverage the structural information for node classification. Despite the good performance on many graphs, GNNs are not able to well utilize the textual information, and thus GNNs often suffer on nodes with few neighbors.

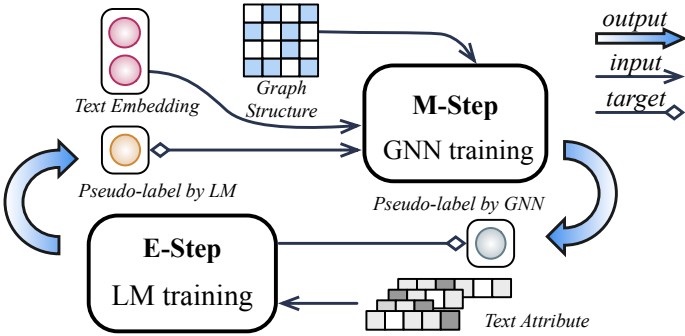

Figure 1: The proposed GLEM framework trains GNN and LM separately in a variational EM framework: In E-step, an LM is trained towards predicting both the gold labels and GNN-predicted pseudo-labels; In M-step, a GNN is trained by predicting both gold labels and LM-inferred pseudo-labels using the embeddings and pseudo-labels predicted by LM.

## 4 METHODOLOGY

In this section, we introduce our proposed approach which combines GNN and LM for node representation learning in TAGs. Existing methods either suffer from scalability issues or have poor results in downstream applications such as node classification. Therefore, we are looking for an approach that enjoys both good scalability and capacity.

Toward this goal, we take node classification as an example and propose GLEM. GLEM leverages a variational EM framework, where the LM uses the text information of each sole node to predict its label, which essentially models the label distribution conditioned on local text attribute; whereas the GNN leverages the text and label information of surrounding nodes for label prediction, which characterizes the global conditional label distribution. The two modules are optimized by alternating between an E-step and an M-step. In the E-step, we fix the GNN and let the LM mimic the labels inferred by the GNN, allowing the global knowledge learned by the GNN to be distilled into the LM. In the M-step, we fix the LM, and the GNN is optimized by using the node representations learned by the LM as features and the node labels inferred by the LM as targets. By doing this, the GNN can effectively capture the global correlations of nodes for precise label prediction. With such a framework, the LM and GNN can be trained separately, leading to better scalability. Meanwhile, the LM and GNN are encouraged to benefit each other, without sacrificing model performance.

### 4.1 THE PSEUDO-LIKELIHOOD VARIATIONAL FRAMEWORK

Our approach is based on a pseudo-likelihood variational framework, which offers a principled and flexible formalization for model design. To be more specific, the framework tries to maximize the log-likelihood function of the observed node labels, i.e., $p(\mathbf{y}_L|\mathbf{s}_V, A)$. Directly optimizing the function is often hard due to the unobserved node labels $\mathbf{y}_U$, and thus the framework instead optimizes the evidence lower bound as below:

$$\log p(\mathbf{y}_L|\mathbf{s}_V, A) \geq \mathbb{E}_{q(\mathbf{y}_U|\mathbf{s}_U)}[\log p(\mathbf{y}_L, \mathbf{y}_U|\mathbf{s}_V, A) - \log q(\mathbf{y}_U|\mathbf{s}_U)], \tag{3}$$

where $q(\mathbf{y}_U|\mathbf{s}_U)$ is a variational distribution and the above inequality holds for any $q$. The ELBO can be optimized by alternating between optimizing the distribution $q$ (i.e., E-step) and the distribution $p$

(i.e., M-step). In the E-step, we aim at updating $q$ to minimize the KL divergence between $q(\mathbf{y}_U|\mathbf{s}_U)$ and $p(\mathbf{y}_U|\mathbf{s}_V, A, \mathbf{y}_L)$, so that the above lower bound can be tightened. In the M-step, we then update $p$ towards maximizing the following pseudo-likelihood (Besag, 1975) function:

$$\mathbb{E}_{q(\mathbf{y}_U|\mathbf{s}_U)}[\log p(\mathbf{y}_L, \mathbf{y}_U|\mathbf{s}_V, A)] \approx \mathbb{E}_{q(\mathbf{y}_U|\mathbf{s}_U)}[\sum_{n \in V} \log p(\mathbf{y}_n|\mathbf{s}_V, A, \mathbf{y}_{V \setminus n})]. \quad (4)$$

The pseudo-likelihood variational framework yields a formalization with two distributions to maximize data likelihood. The two distributions are trained via a separate E-step and M-step, and thus we no longer need the end-to-end training paradigm, leading to better scalability which naturally fits our scenario. Next, we introduce how we apply the framework to node classification in TAGs by instantiating the $p$ and $q$ distributions with GNNs and LMs respectively.

## 4.2 PARAMETERIZATION

The distribution $q$ aims to use the text information $\mathbf{s}_U$ to define node label distribution. In GLEM, we use a mean-field form, assuming the labels of different nodes are independent and the label of each node only depends on its own text information, yielding the following form of factorization:

$$q_\theta(\mathbf{y}_U|\mathbf{s}_U) = \prod_{n \in U} q_\theta(\mathbf{y}_n|\mathbf{s}_n). \quad (5)$$

As introduced in Section 3.2, each term $q_\theta(\mathbf{y}_n|\mathbf{s}_n)$ can be modeled by a transformer-based LM $q_\theta$ parameterized by $\theta$, which effectively models the fine-grained token interactions by the attention mechanism (Vaswani et al., 2017).

On the other hand, the distribution $p$ defines a conditional distribution $p_\phi(\mathbf{y}_n|\mathbf{s}_V, A, \mathbf{y}_{V \setminus n})$, aiming to leverage the node features $\mathbf{s}_V$, graph structure $A$, and other node labels $\mathbf{y}_{V \setminus n}$ to characterize the label distribution of each node $n$. Such a formalization can be naturally captured by a GNN through the message-passing mechanism. Thus, we model $p_\phi(\mathbf{y}_n|\mathbf{s}_V, A, \mathbf{y}_{V \setminus n})$ as a GNN $p_\phi$ parameterized by $\phi$ to effectively model the structural interactions between nodes. Note that the GNN $p_\phi$ takes the node texts $\mathbf{s}_V$ as input to output the node label distribution. However, the node texts are discrete variables, which cannot be directly used by the GNN. Thus, in practice we first encode the node texts with the LM $q_\theta$, and then use the obtained embeddings as a surrogate of node texts for the GNN $p_\phi$.

In the following sections, we further explain how we optimize the LM $q_\theta$ and the GNN $p_\phi$ to let them collaborate with each other.

## 4.3 E-STEP: LM OPTIMIZATION

In the E-step, we fix the GNN and aim to update the LM to maximize the evidence lower bound. By doing this, the global semantic correlations between different nodes can be distilled into the LM.

Formally, maximizing the evidence lower bound with respect to the LM is equivalent to minimizing the KL divergence between the posterior distribution and the variational distribution, i.e., $\mathrm{KL}(q_\theta(\mathbf{y}_U|\mathbf{s}_U)||p_\phi(\mathbf{y}_U|\mathbf{s}_V, A, \mathbf{y}_L))$. However, directly optimizing the KL divergence is nontrivial, as the KL divergence relies on the entropy of $q_\theta(\mathbf{y}_U|\mathbf{s}_U)$, which is hard to deal with. To overcome the challenge, we follow the wake-sleep algorithm (Hinton et al., 1995) to minimize the reverse KL divergence, yielding the following objective function to maximize with respect to the LM $q_\theta$:

$$\begin{aligned} -\mathrm{KL}(p_\phi(\mathbf{y}_U|\mathbf{s}_V, A, \mathbf{y}_L)||q_\theta(\mathbf{y}_U|\mathbf{s}_U)) &= \mathbb{E}_{p_\phi(\mathbf{y}_U|\mathbf{s}_V, A, \mathbf{y}_L)}[\log q_\theta(\mathbf{y}_U|\mathbf{s}_U)] + \mathrm{const} \\ &= \sum_{n \in U} \mathbb{E}_{p_\phi(\mathbf{y}_n|\mathbf{s}_V, A, \mathbf{y}_L)}[\log q_\theta(\mathbf{y}_n|\mathbf{s}_n)] + \mathrm{const}, \end{aligned} \quad (6)$$

which is more tractable as we no longer need to consider the entropy of $q_\theta(\mathbf{y}_U|\mathbf{s}_U)$. Now, the sole difficulty lies in computing the distribution $p_\phi(\mathbf{y}_n|\mathbf{s}_V, A, \mathbf{y}_L)$. Remember that in the original GNN which defines the distribution $p_\phi(\mathbf{y}_n|\mathbf{s}_V, A, \mathbf{y}_{V \setminus n})$, we aim to predict the label distribution of a node $n$ based on the surrounding node labels $\mathbf{y}_{V \setminus n}$. However, in the above distribution $p_\phi(\mathbf{y}_n|\mathbf{s}_V, A, \mathbf{y}_L)$, we only condition on the observed node labels $\mathbf{y}_L$, and the labels of other nodes are unspecified, so we cannot compute the distribution directly with the GNN. In order to solve the problem, we propose

to annotate all the unlabeled nodes in the graph with the pseudo-labels predicted by the LM, so that we can approximate the distribution as follows:

$$p_\phi(\mathbf{y}_n|\mathbf{s}_V, A, \mathbf{y}_L) \approx p_\phi(\mathbf{y}_n|\mathbf{s}_V, A, \mathbf{y}_L, \hat{\mathbf{y}}_{U\setminus n}), \tag{7}$$

where $\hat{\mathbf{y}}_{U\setminus n} = \{\hat{\mathbf{y}}_{n'}\}_{n'\in U\setminus n}$ with each $\hat{\mathbf{y}}_{n'} \sim q_\theta(\mathbf{y}_{n'}|\mathbf{s}_{n'})$.

Besides, the labeled nodes can also be used for training the LM. Combining it with the above objective function, we obtain the final objective function for training the LM:

$$\mathcal{O}(q) = \alpha \sum_{n\in U} \mathbb{E}_{p(\mathbf{y}_n|\mathbf{s}_V, A, \mathbf{y}_L, \hat{\mathbf{y}}_{U\setminus n})}[\log q(\mathbf{y}_n|\mathbf{s}_n)] + (1-\alpha) \sum_{n\in L} \log q(\mathbf{y}_n|\mathbf{s}_n), \tag{8}$$

where $\alpha$ is a hyperparameter. Intuitively, the second term $\sum_{n\in L} \log q(\mathbf{y}_n|\mathbf{s}_n)$ is a supervised objective which uses the given labeled nodes for training. Meanwhile, the first term could be viewed as a knowledge distilling process which teaches the LM by forcing it to predict the label distribution based on neighborhood text-information.

## 4.4 M-STEP: GNN OPTIMIZATION

During the GNN phase, we aim at fixing the language model $q_\theta$ and optimizing the graph neural network $p_\phi$ to maximize the pseudo-likelihood as introduced in equation 4.

To be more specific, we use the language model to generate node representations $\mathbf{h}_V$ for all nodes and feed them into the graph neural network as text features for message passing. Besides, note that equation 4 relies on the expectation with respect to $q_\theta$, which can be approximated by drawing a sample $\hat{\mathbf{y}}_U$ from $q_\theta(\mathbf{y}_U|\mathbf{s}_U)$. In other words, we use the language model $q_\theta$ to predict a pseudo-label $\hat{\mathbf{y}}_n$ for each unlabeled node $n \in U$, and combine all the labels $\{\hat{\mathbf{y}}_n\}_{n\in U}$ into $\hat{\mathbf{y}}_U$. With both the node representations and pseudo-labels from the LM $q_\theta$, the pseudo-likelihood can be rewritten as follows:

$$\mathcal{O}(\phi) = \beta \sum_{n\in U} \log p_\phi(\hat{\mathbf{y}}_n|\mathbf{s}_V, A, \mathbf{y}_L, \hat{\mathbf{y}}_{U\setminus n}) + (1-\beta) \sum_{n\in L} \log p_\phi(\mathbf{y}_n|\mathbf{s}_V, A, \mathbf{y}_{L\setminus n}, \hat{\mathbf{y}}_U), \tag{9}$$

where $\beta$ is a hyperparameter which is added to balance the weight of the two terms. Again, the first term can be viewed as a knowledge distillation process which injects the knowledge captured by the LM into the GNN via all the pseudo-labels. The second term is simply a supervised loss, where we use observed node labels for model training.

Finally, the workflow of the EM algorithm is summarized in Fig. 1. The optimization process iteratively does the E-step and the M-step. In the E-step, the pseudo-labels predicted by the GNN together with the observed labels are utilized for LM training. In the M-step, the LM provides both text embeddings and pseudo-labels for the GNN, which are treated as input and target respectively for label prediction. Once trained, both the LM in E-step (denoted as GLEM-LM) and the GNN (denoted as GLEM-GNN) in M-step can be used for node label prediction.

## 5 EXPERIMENTS

In this section, we conduct experiments to evaluate the proposed GLEM framework, where two settings are considered. The first setting is transductive node classification, where given a few labeled nodes in a TAG, we aim to classify the rest of the nodes. Besides that, we also consider a structure-free inductive setting, and the goal is to transfer models trained on labeled nodes to unseen nodes, for which we only observe the text attributes without knowing their connected neighbors.

### 5.1 EXPERIMENTAL SETUP

**Datasets.** Three TAG node classification benchmarks are used in our experiment, including ogbn-arxiv, ogbn-products, and ogbn-papers100M (Hu et al., 2020). The statistics of these datasets are shown in Table 1.

**Compared Methods.** We compare GLEM-LM and GLEM-GNN against LMs, GNNs, and methods combining both of worlds. For language models, we apply DeBERTa He et al. (2021) to our

Table 1: Statistics of the OGB datasets (Hu et al., 2020).

|  | #Nodes | #Edges | Avg. Node Degree | Train / Val / Test (%) |
|---|---|---|---|---|
| ogbn-arxiv (Arxiv) | 169,343 | 1,166,243 | 13.7 | 54 / 18 / 28 |
| ogbn-products (Products) | 2,449,029 | 61,859,140 | 50.5 | 8 / 2 / 90 |
| ogbn-papers100M (Papers) | 111,059,956 | 1,615,685,872 | 29.1 | 78 / 8 / 14 |

Table 2: Node classification accuracy for the Arxiv and Products datasets. (mean ± std%, the best results are bolded and the runner-ups are underlined). G ↑ denotes the improvements of GLEM-GNN over the same GNN trained on $\mathbf{X}_{\text{OGB}}$; L ↑ denotes the improvements of GLEM-LM over LM-Ft. "+" denotes additional tricks are implemented in the original GNN models.

| Datasets | Methods | | GNN | | | | | LM | | |
|---|---|---|---|---|---|---|---|---|---|---|
| | | | $\mathbf{X}_{\text{OGB}}$ | $\mathbf{X}_{\text{GIANT}}$ | $\mathbf{X}_{\text{PLM}}$ | GLEM-GNN | G ↑ | LM-Ft | GLEM-LM | L↑ |
| Arxiv | GCN | val | 73.00 ± 0.17 | 74.89 ± 0.17 | 47.56 ± 1.91 | 76.86 ± 0.19 | 3.86 | 75.27 ± 0.09 | 76.17 ± 0.47 | 0.90 |
| | | test | 71.74 ± 0.29 | 73.29 ± 0.10 | 48.19 ± 1.47 | 75.93 ± 0.19 | 4.19 | 74.13 ± 0.04 | 75.71 ± 0.24 | 1.58 |
| | SAGE | val | 72.77 ± 0.16 | 75.95 ± 0.11 | 56.16 ± 0.46 | 76.45 ± 0.05 | 3.68 | 75.27 ± 0.09 | 75.32 ± 0.04 | 0.6 |
| | | test | 71.49 ± 0.27 | 74.35 ± 0.14 | 56.39 ± 0.82 | 75.50 ± 0.24 | 4.01 | 74.13 ± 0.04 | 74.53 ± 0.12 | 1.44 |
| | GAMLP | val | 62.20 ± 0.11 | 75.01 ± 0.02 | 71.14 ± 0.19 | 76.95 ± 0.14 | 14.75 | 75.27 ± 0.09 | 75.64 ± 0.30 | 0.44 |
| | | test | 56.53 ± 0.02 | 73.35 ± 0.14 | 70.15 ± 0.22 | 75.62 ± 0.23 | 19.09 | 74.13 ± 0.04 | 74.48 ± 0.41 | 2.04 |
| | RevGAT | val | 75.01 ± 0.10 | 77.01 ± 0.09 | 71.40 ± 0.23 | 77.49 ± 0.17 | 2.48 | 75.27 ± 0.09 | 75.75 ± 0.07 | 0.48 |
| | | test | 74.02 ± 0.18 | 75.90 ± 0.19 | 70.21 ± 0.30 | **76.97 ± 0.19** | 2.95 | 74.13 ± 0.04 | 75.45 ± 0.12 | 1.32 |
| Products | SAGE | val | 91.99 ± 0.07 | 93.47 ± 0.14 | 86.74 ± 0.31 | 93.84 ± 0.12 | 1.85 | 91.82 ± 0.11 | 92.71 ± 0.15 | 0.71 |
| | | test | 79.21 ± 0.15 | 82.33 ± 0.37 | 71.09 ± 0.65 | 83.16 ± 0.19 | 3.95 | 79.63 ± 0.12 | 81.25 ± 0.15 | 1.61 |
| | GAMLP | val | 93.12 ± 0.03 | 93.99 ± 0.04 | 91.65 ± 0.17 | 94.19 ± 0.01 | 1.07 | 91.82 ± 0.11 | 90.56 ± 0.04 | -1.26 |
| | | test | 83.54 ± 0.09 | 83.16 ± 0.07 | 80.49 ± 0.19 | 85.09 ± 0.21 | 1.55 | 79.63 ± 0.12 | 82.23 ± 0.27 | 2.60 |
| | SAGN+ | val | 93.02 ± 0.04 | 93.64 ± 0.05 | 92.78 ± 0.04 | 94.00 ± 0.03 | 0.98 | 91.82 ± 0.11 | 92.01 ± 0.05 | 0.21 |
| | | test | 84.35 ± 0.09 | 86.67 ± 0.09 | 84.20 ± 0.39 | **87.36 ± 0.07** | 3.01 | 79.63 ± 0.12 | 84.83 ± 0.04 | 5.17 |
| Papers | GAMLP | val | 71.17 ± 0.14 | 72.70 ± 0.07 | 69.78 ± 0.07 | 71.71 ± 0.09 | 0.54 | 68.05 ± 0.03 | 69.94 ± 0.16 | 1.89 |
| | | test | 67.71 ± 0.20 | 69.33 ± 0.06 | 65.94 ± 0.10 | 68.25 ± 0.14 | 0.54 | 63.52 ± 0.06 | 64.80 ± 0.06 | 1.78 |
| | GAMLP+ | val | 71.59 ± 0.05 | 73.05 ± 0.04 | 69.87 ± 0.06 | 73.54 ± 0.01 | 1.95 | 68.05 ± 0.03 | 71.16 ± 0.45 | 3.11 |
| | | test | 68.25 ± 0.11 | 69.67 ± 0.05 | 66.36 ± 0.09 | **70.36 ± 0.02** | 2.11 | 63.52 ± 0.06 | 66.71 ± 0.25 | 3.19 |

setting by fine-tuning it on labeled nodes, and we denote it as LM-Ft. For GNNs, a few well-known GNNs are selected, i.e., GCN (Kipf & Welling, 2017) and GraphSAGE (Hamilton et al., 2017). Three top-ranked baselines on leaderboards are included, i.e., RevGAT (Li et al., 2021b), GAMLP (Zhang et al., 2022), SAGN (Sun & Wu, 2021). For each GNN, we try different kinds of node features, including (1) the raw feature of OGB, denoted as $\mathbf{X}_{\text{OGB}}$; (2) the LM embedding inferenced by pre-trained LM, i.e. the DeBERTa-base checkpoint [2], denoted as $\mathbf{X}_{\text{PLM}}$; (3) the GI-ANT (Chien et al., 2022) feature, denoted as $\mathbf{X}_{\text{GIANT}}$.

**Implementation Details.** We adopt the DeBERTa (He et al., 2021) as the LM model and fine-tune it for node classification to provide initial checkpoints for LMs and further infer text embeddings and predictions for the first GNN M-step. To provide predictions for the first-LM E-step, we use pre-trained GNN predictions, e.g. the original GNN predictions, for the initial target labels. The best EM-iteration is chosen based on the validation accuracy of GLEM-GNN. During optimization, GLEM can start with either the E-step or the M-step. For better performance, we let the better module generates pseudo-labels and train the other module first. For example, if pre-trained GNN outperforms pre-trained LM, we start with the E-step (LM training). For fair comparison against other feature learning methods such as GIANT, the hyper-parameters of GNNs are set to the best settings described in the paper or in the official repository, other parameters are tuned by grid search.

## 5.2 Transductive Node Classification

**Main Results.** Next, we evaluate GLEM in the transductive setting. The results of three OGB datasets are presented in Table 2. For LMs, we see that fine-tuned LMs (LM-Ft) have competitive results, showing the importance of text attributes in a TAG. By further leveraging the structural information for message passing, our approach (GLEM-LM) achieves significant improvement over LMs, which demonstrates its advantage over LMs.

For GNN-based methods, we see that for each GNN model, using OGB and GIANT node embeddings as GNN inputs ($\mathbf{X}_{\text{OGB}}$ and $\mathbf{X}_{\text{GIANT}}$) yields strong results. However, these embeddings remain unchanged during training. By dynamically updating the LM to generate more useful node embed-

---
[2]https://huggingface.co/microsoft/DeBERTa-base

Table 3: Experiments on Arxiv (RevGAT as GNN backbone) with different scale of LM.

| Methods | Val. accuracy | Test accuracy | # Parameters |
|---|---|---|---|
| GNN-$\mathbf{X}_{\text{OGB}}$ | 75.01 ± 0.10 | 74.02 ± 0.18 | 2,098,256 |
| GNN-$\mathbf{X}_{\text{GIANT}}$ | 77.01 ± 0.09 | 75.90 ± 0.19 | 1,304,912 |
| GLEM-GNN-base | 77.49 ± 0.17 | 76.97 ± 0.19 | 1,835,600 |
| GLEM-GNN-large | 77.92 ± 0.06 | 77.62 ± 0.16 | 2,228,816 |
| LM-base-Ft | 75.27 ± 0.09 | 74.13 ± 0.04 | 138,632,488 |
| LM-large-Ft | 75.08 ± 0.06 | 73.81 ± 0.08 | 405,204,008 |
| GLEM-LM-base | 75.75 ± 0.07 | 75.45 ± 0.12 | 138,632,488 |
| GLEM-LM-large | 77.16 ± 0.04 | 76.80 ± 0.05 | 405,204,008 |

Table 4: Structure free inductive experiments on ogbn-arxiv and ogbn-products. The validation and test accuracies, denoted as w/ struct and wo struct, and their relative differences (the lower the better) are reported. Boldfaced numbers indicate the best performances in each group.

| Type | Methods | Arxiv | | | Products | | |
|---|---|---|---|---|---|---|---|
| | | w/ struct | wo struct | diff | w/ struct | wo struct | diff |
| MLP | $\mathbf{X}_{\text{OGB}}$ | 57.65 ± 0.12 | 55.50 ± 0.23 | -2.15 | 75.54 ± 0.14 | 61.06 ± 0.08 | -14.48 |
| | $\mathbf{X}_{\text{LM-Ft}}$ | 74.56 ± 0.01 | 72.98 ± 0.06 | **-1.58** | 91.79 ± 0.01 | 79.93 ± 0.22 | -11.86 |
| | $\mathbf{X}_{\text{GLEM}}$-light | 75.20 ± 0.03 | 73.32 ± 0.31 | -1.88 | 91.96 ± 0.01 | 79.38 ± 0.14 | -12.58 |
| | $\mathbf{X}_{\text{GLEM}}$-deep | 75.57 ± 0.03 | **73.90 ± 0.08** | -1.67 | 91.85 ± 0.02 | **80.04 ± 0.15** | **-11.81** |
| GNN | $\mathbf{X}_{\text{OGB}}$-light | 70.73 ± 0.02 | 48.59 ± 0.19 | -22.14 | 90.54 ± 0.04 | 51.23 ± 0.17 | -39.31 |
| | $\mathbf{X}_{\text{GLEM}}$-light | 76.73 ± 0.02 | 73.94 ± 0.03 | -2.79 | 92.95 ± 0.03 | 78.75 ± 0.39 | -14.20 |
| | $\mathbf{X}_{\text{OGB}}$-deep | 72.67 ± 0.03 | 50.92 ± 0.19 | -21.75 | 91.85 ± 0.11 | 32.71 ± 2.23 | -59.14 |
| | $\mathbf{X}_{\text{GLEM}}$-deep | 76.79 ± 0.06 | **74.29 ± 0.11** | **-2.50** | 93.22 ± 0.03 | **79.81 ± 0.01** | **-13.41** |
| LM | Fine-tune | 75.27 ± 0.09 | 74.13 ± 0.04 | -1.14 | 91.82 ± 0.11 | 79.63 ± 0.12 | -12.19 |
| | GLEM-light | 75.49 ± 0.11 | 74.50 ± 0.16 | -0.99 | 91.90 ± 0.06 | 79.53 ± 0.13 | -12.37 |
| | GLEM-deep | 75.59 ± 0.08 | **74.60 ± 0.05** | **-0.99** | 91.81 ± 0.04 | **79.69 ± 0.51** | **-12.12** |

dings and pseudo-labels for the GNN, GLEM-GNN significantly outperforms all the other methods with fixed node embeddings in most cases. Notably, GLEM-GNN achieves new state-of-the-art results on all three TAG datasets on the OGB benchmark.

**Scalability.** One key challenge of fusing LMs and GNNs lies in scalability. When using large LMs with numerous parameters, the combined method will suffer from severe scalability problems. GLEM eases this challenge through the EM-based optimization paradigm, allowing it to be adapted to large LMs. To verify this claim, we train GLEM with DeBERTa-large (He et al., 2021) on ogbn-arxiv. The results are reported in Table 3. We observe that GLEM is able to generalize to DeBERTa-large with about 0.4B parameters, showing the appealing scalability. Besides, for every LM, applying GLEM yields consistent improvement, which proves the effectiveness of GLEM.

### 5.3 STRUCTURE-FREE INDUCTIVE NODE CLASSIFICATION

Besides the transductive setting, inductive settings are also important, where we aim to train models on nodes of a training graph, and then generalize models to unobserved nodes. In many real cases, these new nodes are often low-degree nodes or even isolated nodes, meaning that we can hardly use the structural information for node classification. Therefore, we consider a challenging setting named structure-free inductive setting, where we assume for each test node, we only observe its text attributes without any connected neighbors. For this setting, we consider different types of methods for label prediction, including GNN models (GNN), neural networks without using structural information (MLP), and GLEM. The results are shown in Table 4.

We can see that the structure-free inductive setting is a more challenging task, especially for GNNs where a sheer performance drop is observed. Meanwhile, by effectively fusing with graph learning, GLEM-LM is able to consider local semantics as well as neighboring structural information, leading to more accurate structure-free inductive predictions. Besides, the generated embeddings are able to boost other models (e.g., see $\mathbf{X}_{\text{GLEM}}$-deep in the MLP and LM sections), enabling both MLP and GNNs with better structure-free inference ability.

Table 5: Comparison of different training paradigms of fusing LM and GNNs. The maximum batch size (max bsz.) and time/epoch are tested on a single NVIDIA Tesla V100 32GB GPU.

| Datasets | Metric | LM-Ft DeBERTa-base | Static SAGE-$X_{OGB}$ | Joint joint-BERT-tiny | Joint GraphFormers | GLEM GLEM-GNN | GLEM GLEM-LM |
|---|---|---|---|---|---|---|---|
| Arxiv | val. acc. | 75.27 ± 0.09 | 72.77 ± 0.16 | 71.58 ± 0.18 | 73.33 ± 0.06 | 76.45 ± 0.05 | 75.32 ± 0.04 |
| | test acc. | 74.13 ± 0.04 | 71.49 ± 0.27 | 70.87 ± 0.12 | 72.81 ± 0.20 | 75.50 ± 0.24 | 74.53 ± 0.12 |
| | parameters | 138,632,488 | 218,664 | 110,694,592 | 110,694,592 | 545,320 | 138,632,488 |
| | max bsz. | 30 | all nodes | 200 | 180 | all nodes | 30 |
| | time/epoch | 2760s | 0.09s | 1827s | 4824s | 0.13s | 3801s |
| Products | val. acc. | 91.82 ± 0.11 | 91.99 ± 0.07 | 90.85 ± 0.12 | 91.77 ± 0.09 | 93.84 ± 0.12 | 92.71 ± 0.15 |
| | test acc.. | 79.63 ± 0.12 | 79.21 ± 0.15 | 73.13 ± 0.11 | 74.72 ± 0.16 | 83.16 ± 0.19 | 81.25 ± 0.15 |
| | parameters | 138,637,871 | 206,895 | 110,699,975 | 110,699,975 | 548,911 | 138,637,871 |
| | max bsz. | 30 | all nodes | 100 | 100 | 80000 | 30 |
| | time/epoch | 5460s | 8.1s | 8456s | 12574s | 153s | 7740s |

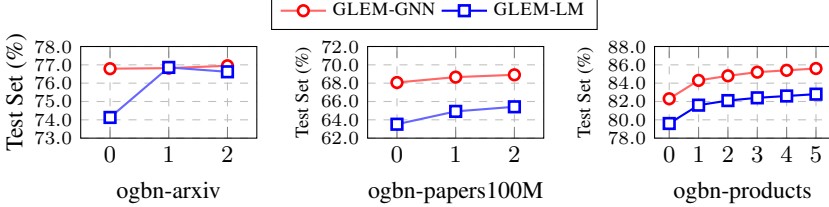

Figure 2: The convergence curves of GLEM on OGB datasets.

## 5.4 COMPARISON OF DIFFERENT TRAINING PARADIGMS

As discussed before, besides directly fine-tuning LM (denoted as **LM-Ft**), a few training paradigms have been proposed for fusing GNNs and LMs. One paradigm is to use fixed/static LMs to generate node embeddings for GNN to do label prediction (denoted as **Static**). Besides that, another paradigm is to restrict message passing to a few sampled neighbors, so that the memory cost can be reduced (denoted as **Joint**). In this section, we compare our proposed paradigm (**GLEM**) against the others. For each paradigm, we choose two models trained with it. The results are presented in Table 5. we see that although static training has the best efficiency, its classification accuracy is not very high due to the restricted model capacity caused by the fixed LM. On the other hand, the joint training paradigm has the worst efficiency and effectiveness due to the reduced graph structure. Finally, our proposed paradigm achieves the optimal classification results, thanks to its ability to encourage the collaboration of LMs and GNNs. Meanwhile, our paradigm remains close to static training in terms of efficiency (time/epoch). To summarize, our proposed approach achieves much better results than other paradigms without sacrificing efficiency.

## 5.5 CONVERGENCE ANALYSIS

In GLEM, we utilize the variational EM algorithm for optimization, which consists of an E-step training GLEM-LM and an M-step training GLEM-GNN in each iteration. Here, we analyze the convergence of GLEM by looking into the training curves of validation accuracy on ogbn-arxiv and OGB-Products. From the results in Fig. 2. We can clearly see that with each E-step and M-step, both the performance of GLEM-GNN and the GLEM-LM consistently increase to a maximum point and converge in a few iterations. Notably, GLEM takes only one iteration to converge on the ogbn-arxiv dataset, which is very efficient.

## 6 CONCLUSION

This paper studies how to fuse LMs and GNNs together for node representation learning in TAGs. We propose an approach GLEM based on a pseudo-likelihood variational framework. GLEM alternatively updates the LM and GNN via an E-step and an M-step, allowing for better scalability. In each step, both GNN and LM are mutually enhanced by learning from pseudo-labels predicted by the other module, fusing graph and language learning together. Extensive experiments on multiple datasets in two settings demonstrate the effectiveness and efficiency of GLEM.

ACKNOWLEDGEMENT

This project is supported by Twitter, Intel, the Microsoft Research Asia internship program, the Natural Sciences and Engineering Research Council (NSERC) Discovery Grant, the Canada CIFAR AI Chair Program, collaboration grants between Microsoft Research and Mila, Samsung Electronics Co., Ltd., Amazon Faculty Research Award, Tencent AI Lab Rhino-Bird Gift Fund, a NRC Collaborative R&D Project (AI4D-CORE-06) as well as the IVADO Fundamental Research Project grant PRF-2019-3583139727.

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

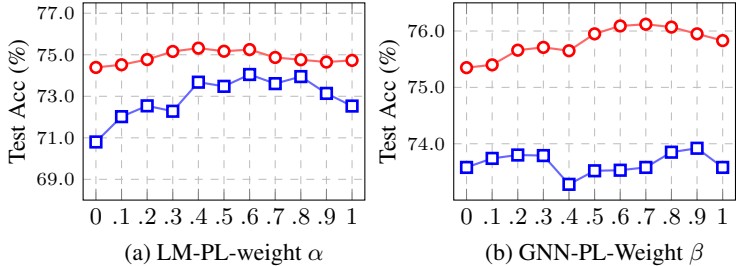

Figure 3: The effect of the $\alpha$ and $\beta$ for GLEM-GCN.

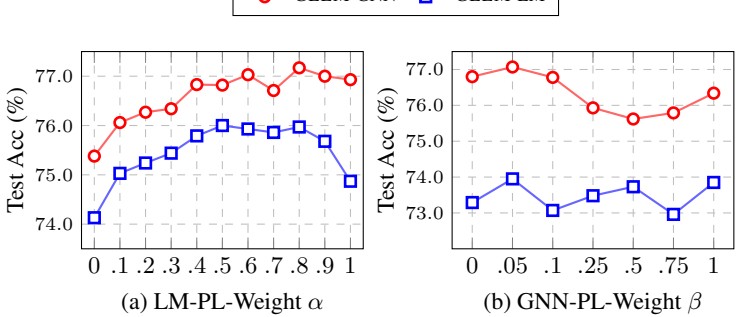

Figure 4: The effect of the $\alpha$ and $\beta$ for GLEM-RevGAT.

## A   SENSITIVITY ANALYSIS

Recall that GLEM fuses LMs and GNNs by training them separately with an E-step and an M-step, where at each step both the observed labels and pseudo-labels are used for model training as shown in Eq.8 and Eq.9. For both objectives, we introduce a coefficient, i.e., $\alpha$ and $\beta$) denoted as LM-PL-weight and GNN-PL-weight respectlively, to control the relative weight of pseudo-labels. Next, we systematically analyze these two hyperparameters by investigating how the performance varies when changing the hyperparameters. We treat GCN (Kipf & Welling, 2017) and RevGAT (Li et al., 2021b) as the backbone GNNs. The results on the OGBN-Arxiv dataset are shown in Figure 3 and Figure 4 respectively.

On one hand, we can clearly see that the LM-PL-weight $\alpha$ is an important parameter, as GLEM-LM provides both node feature and pseudo-labels for GLEM-GNN. We also observe that guiding LM with pseudo labels of GNN consistently boosts the performance of both LM and GNN compared with optimizing LM with gold label only, i.e. $\alpha = 0$, demonstrating the effectiveness of fusing the knowledge of GNN into LM. On the other hand, the GNN-PL-weight $\alpha$ that balance the important of LM pseudo-labels in training GLEM-GNN is not very sensitive. Lastly, we also observe that for different GNN and LM, the optimal $\alpha$ and $\beta$ varies, indicating these parameters should be carefully selected.

## B   REPRODUCIBILITY STATEMENT

We provide our code in a public repository along with detailed instructions on conducting experiments. Our experimental settings and implementation details are stated in Section 5.1, the important hyper-parameters are discussed in the Appendix.

