# OpenReview forum: "Learning on Large-scale Text-attributed Graphs via Variational Inference"
_ICLR.cc/2023/Conference — ICLR 2023 notable top 5%_

### Official Review · Reviewer_oKS1 · 2022-10-21

**Confidence:** 3
**Correctness:** 3
**Technical Novelty And Significance:** 3
**Empirical Novelty And Significance:** 3
**Recommendation:** 6

**Clarity, Quality, Novelty And Reproducibility:**

Clarity issues/questions:

1. More details of the implementation are needed, including the detailed architecture, learning rates, optimizers, ...

2. What's the proportion of the node with/without labels?

3. What's the model for the implementation of the right-hand-side term of Eq (7)?

4. What's the difference between GAMLP and GAMLP+?

Minor clarity issues:

1. SAGN+ in Table 2 is not defined.


2. A typo in Eq (6)

3. In Table 1, the train/val/test proportions are 8/2/90?

Without the details of the implementation or the release of the code, I have concerns to the reproducibility of the paper.


**Strength And Weaknesses:**

Strength:

1. The general idea of this paper is performing joint training for LM and GNN and letting them help each other during training. This might be done with heuristic approaches. But in this paper, it is interesting to wrap up the alternative training of LM and GNN into an EM framework, which adds credit to the technical depth of the paper.

2. The datasets and the baselines are well selected and the experiments are comprehensive in general including different settings and comprehensive ablation studies.

Weaknesses:


1. Some of the performance improvement of the proposed method is a bit marginal. For example, in Table 2, GLEM-GNN (the proposed method) is marginally better than the second best (X_{GIANT}) on arxiv and products (e.g., 75.90 ± 0.19 VS 76.74 ± 0.08 with RevGAT), not mentioning that GLEM-GNN did not outperform X_{GIANT} on papers.

2. Several clarity issues (please see comments in the next section).


**Summary Of The Paper:**

This paper presents a new method for node classification where nodes are associated with text attributes. The proposed framework consists of a language model (LM) and a graph neural network (GNN) and uses the variational EM algorithm that learns LM and GNN in the E and M steps respectively. The proposed method is evaluated on several large-scale graphs with text attributes in comparison with different LMs and GNNs used separately.

**Summary Of The Review:**

The variational EM formulation is interesting but the quality of the paper needs to be improved.

The author provided detailed information to my concerns of clarity. I've increased my rating.

---

> ### Author Response · Authors · 2022-11-12
> **Response to Reviewer oKS1**
>
> We sincerely appreciate your kind comments and your positive assessment. We hope our point-to-point response can address your concerns.
>
> ### Response to the Proposed Weaknesses
>
> **W1. About the model performance.**
>
> As you pointed out, our approach was inferior to X-GIANT on the Papers dataset. This was because Papers is the biggest dataset and we did not spend enough time tuning the hyperparameters before submission. Recently, we did more experiments, and we presented the results as follows:
>
> |                                                | Arxiv                              | Products                       | Papers                      |
> |------------------------------------------------|------------------------------------|--------------------------------|-----------------------------|
> | Top1 model/test acc. %                         | GLEM-RevGAT / 76.94                | GLEM-EnGCN /90.14              | GLEM-GAMLP / 70.36          |
> | Top2 model/test acc. %                         | GIANT-XRT+AGDN+BoT+self-KD / 76.37 | EnGCN / 87.98                  | GIANT-XRT+GAMLP+RLU / 69.67 |
> | GIANT (Nov2021) model/test acc. %              | GIANT-XRT+RevGAT+KD/ 76.15         | GIANT-XRT+SAGN+SLE+C&S / 86.43 | GIANT-XRT+GAMLP+RLU / 69.67 |
> | Improvement of GLEM over the second-best model | 0.57%                              | 2.16%                          | 0.69%                       |
> | Leaderboard Imp. from Nov2021 to Sep2022       | 0.22%                              | 0.41%                          | 0%                          |
>
> From the results, we can observe:
> (1) Our approach now consistently outperforms X-GIANT.
> (2) GLEM now achieves new SOTA performance on all TAG datasets of the OGBN benchmark, surpassing the second best with 0.57%, 2.16%, and 0.69% improvement. In comparison, from Nov2021 (date of GIANT's release) to Sep2022 (ICLR Deadline), the performance on these datasets was only improved by 0.22%, 0.41%, and 0%.
> Based on the above observations, we would kindly say that the improvement made by our approach is quite significant.
>
> **W2. Regarding the clarity issues.**
>
> Q1. Details of implementation & Reproducibility
>
> C1. We are sorry we missed the reproducibility statement, and thank you for pointing it out! We have updated the reproducibility statement in the revised version. We also uploaded the code in the supplementary materials. We hope these materials may ease your concern about reproducibility.
>
> Q2. The proportion of nodes with/without labels
>
> C2. On Arxiv and Products, all nodes are labeled. On Papers datasets, about 1.5 million of the nodes are labeled. The splits of labeled nodes are shown in Table 1. More information about the data could be found at https://ogb.stanford.edu/docs/nodeprop/.
>
> Q3. What's the model for the implementation of the right-hand-side term of Eq (7)?
>
> C3. We are sorry about the confusion. The right-hand-side term is the label distribution predicted by the GNN. More specifically, we feed node sentences, edges between nodes, and node labels into a GNN, which further outputs the distribution of node n’s label.
>
> Q4. Clarification of GAMLP+ and SAGN+
>
> C4. We are sorry for the confusion. With the fierce competition on the OGB leaderboard, SOTA methods usually introduce several tricks in the model. Take SAGN as an example, it utilizes the X-GIANT, SCR, C&S tricks, the official name of the model on the leaderboard is denoted as GIANT-XRT+SAGN+SCR+C&S, which is extremely long and takes a huge space in the table. For RLU, the official full name is “GIANT-XRT+GAMLP+RLU (use raw text)”. For aesthetics, we use a mark “+” behind the model name to denote that additional tricks are implemented in the original GNN models. Please kindly note that these tricks are proposed and implemented by the original code of GAMLP/SAGN on the leaderboard.
>
> Q5. In Table 1, the train/val/test proportions are 8/2/90?
>
> C5. Yes. This is [the official split](https://ogb.stanford.edu/docs/nodeprop/#ogbn-products) of the OGB leaderboard. The split is conducted by: “The products are sorted according to their sales ranking and use the top 8% for training, next top 2% for validation, and the rest for testing. This is a more challenging splitting procedure that closely matches the real-world application where labels are first assigned to important nodes in the network and ML models are subsequently used to make predictions on less important ones.”
>
> ### Revision of Paper
>
> We sincerely appreciate your suggestions and fixed the typos in the revised paper.

---

### Official Review · Reviewer_qotH · 2022-10-24

**Confidence:** 4
**Correctness:** 3
**Technical Novelty And Significance:** 3
**Empirical Novelty And Significance:** 4
**Recommendation:** 8

**Clarity, Quality, Novelty And Reproducibility:**

This paper is well-written and quite novel. I encourage the authors to open-source the code to improve its reproducibility and impact

**Strength And Weaknesses:**

**Strength**

**Clear motivation**: this paper is clearly motivated to target on the problem of joint learning LM and GNN efficiently.

**Mathematically principled**: the variational EM framework is elegant and very smart. Although variational EM is very well studied and already used for learning GNNs, I think the application of variational EM in this paper is still novel.

**Solid experiment**: the experiment is conducted well, with solid comparison with GNNs and LMs and joint methods.The improvements are also clearly marked in Table 2.

**Efficiency and scalability**: The efficiency (quick enough training time) and scalability (setting larger batch size) is also clear in table 5.

**Weakness**

**Multiple approximation happens in training:** the authors have clearly discussed all the places where approximation happens, which on itself is a good thing. Yet all these approximations (mean-field, weak-sleep, pseudo label, balancing parameters .etc) intuitively makes the final objective deviate very much from the original variational bound (or not sure if the final objective is a bound anymore). Although being orthodoxical to the original variational objective may not be practically good, it would still be mathematically more principled if the authors could establish more rigid analysis of the final objective (e.g., its relationship to the likelihood).

**Summary Of The Paper:**

This paper proposes a variational expectation maximization framework to jointly train the language model and the graph neural network for representation learning for text-attributed graphs. The approach is clearly motivated, mathematically principled, and empirically effective and efficient. I believe this paper makes a clear contribution to graph representation learning and opens new possibilities for scaling graph representation learning to even larger graphs and language models.

**Summary Of The Review:**

This paper proposes a variational expectation maximization framework to jointly train the language model and the graph neural network for representation learning for text-attributed graphs. The method novel and smart. Experiments demonstrate its effectiveness, efficiency and scalability. I believe this paper would make a valid contribution to the community.

---

> ### Author Response · Authors · 2022-11-12
> **Response to Reviewer qotH**
>
> Thank you for the insightful comments! You are correct that GLEM introduces a few approximations, and there are mainly two underlying reasons.
>
> 1. The vanilla likelihood function is hard to be optimized, especially when the graphical structure between nodes is highly complicated. To address the challenge, some approximation methods are often needed, such as pseodolikelihood and the wake-sleep algorithm, and these methods have been proven useful in many early works despite worse theoretical guarantees.
> 2. The major goal of GLEM is to apply the variational framework to large-scale text-attributed graphs for effective node classification. In order to make GLEM scalable with good classification results, we find that some approximation methods are practically helpful (e.g., pseudo labels), so we finally use them, with a sacrifice of rigorousness.
>
> Despite the above reasons, we totally agree with you that it is necessary to establish a more rigid analysis of the objective, and we leave the analysis as a future work.

---

### Official Review · Reviewer_5Tau · 2022-10-25

**Confidence:** 3
**Correctness:** 3
**Technical Novelty And Significance:** 4
**Empirical Novelty And Significance:** 4
**Recommendation:** 8

**Clarity, Quality, Novelty And Reproducibility:**

### Clarity
The overall method and experimental setup are clearly written. More clarifications on implementation details are still needed.

### Quality
The quality of the paper is good.

### Novelty
The method is novel enough.

### Reproducibility
I cannot find the reproducibility statement in the main manuscript. The authors do not provide any codes or software to reproduce their results.
More clarifications on implementation details are needed for reproducibility.

**Strength And Weaknesses:**

### Strengths

- **Simple but powerful idea;** the authors prove that their method outperforms previous baselines on three benchmark datasets. This work has enough impact on the field of representation learning on TAG.

### Weaknesses

- **Lack of ablation studies;** In E-step and M-step, there are hyperparameters that balance the weight of two terms. However, I fail to find any design choice or ablation study on this. In addition, more details (optimizer, learning rate, batch size, …) should be described in the paper (at least appendix) to fully reproduce the methods and experiments in this work.

### Suggestions

- Questions

    - What is the difference between GLEM-LM and GLEM-GNN? Do they differ based on the starting point of optimization (E-step or M-step first)?

- Typos & suggestions
    - Table 3: Dataset name (ogbg-papers) is missing in Table 3.
    - Table 4: In Arxiv-MLP experiments, the boldface on diff is wrong. (on -1.67 instead of -1.58

**Summary Of The Paper:**

In this paper, the authors propose the node representation method GLEM for Text-Attributed Graphs (TAG) based on a pseudo-likelihood variational framework.

Specifically, they alternatively update GNN (M-step) and LM (E-step) with pseudo-labels from each other.

In experiments, they validate their method achieves SOTA on large-scale TAG while maintaining a high level of efficiency.

**Summary Of The Review:**

To my knowledge, this paper has sufficient merits to be accepted.

For reproducibility, there should be more information provided about implementation and experimental settings.

Please be aware that since I am unfamiliar with the works on representation learning of TAGs, I may have missed significant prior works.

---

> ### Author Response · Authors · 2022-11-12
> **Response to Reviewer 5Tau**
>
> We sincerely appreciate your kind comments and your positive assessment. We hope our point-to-point response can address your concerns.
>
> **Weakness: Lack of ablation studies**
>
> We sincerely appreciate your constructive suggestions and added an ablation study regarding this.
> The reproducibility statement is also included in the revised version along with the code in the supplementary materials. Hope this would ease your concern about reproducibility.
>
> **Clarification of the GLEM framework**
>
> Definitions of GLEM-LM and GLEM-GNN: The GLEM framework alternates between an E-step that trains the LM  (denoted as GLEM-LM) and an M-step that trains GNN (denoted as GLEM-GNN). Once trained, both GLEM-GNN and GLEM-LM can be used for node label prediction.
> Order of EM: Just like in the normal EM algorithm, one can start with either the E-step or the M-step. However, as the performance of GNNs and LM differs, we empirically found that letting the better model provide pseudo labels yields better performance. E.g. if the GNN performs better than the LM, then the LM (E-step) should be first trained.
>
> **Revision of Paper**
>
> We sincerely appreciate your suggestions and fixed the typos in the revised paper.

---

### Official Review · Reviewer_5qjT · 2022-10-27

**Confidence:** 4
**Correctness:** 4
**Technical Novelty And Significance:** 4
**Empirical Novelty And Significance:** 3
**Recommendation:** 8

**Clarity, Quality, Novelty And Reproducibility:**

Overall, the paper is well-written, and most of the different concepts are clearly presented. At the same time, the paper keeps a good balance between theoretical contributions and empirical evaluation.

Below, I list a few other points that could improve the clarity and presentation of the paper:
* In the presentation of GNNs for node classification in Sec. 3, it is not clear what are the initial node feature vectors. This becomes more clear afterward but would be helpful to clarify it here as well.
* In the experiments, two variants of GLEM, namely GLEM-GNN and GLEM-LM are used. These models have not been formally defined. I assume each one of them corresponds to the predictions made by the GNN and LM part respectively of the overall pipeline.


**Strength And Weaknesses:**

**Strengths:**
* Interesting formulation of GLEM. Jointly optimizing the LM and the GNN within a variational EM algorithm constitutes an elegant formulation of the problem.

* The paper has performed experiments on large-scale datasets.

**Weaknesses:**
* The paper has mostly examined datasets that contain textual features. However, similar settings arise while dealing with the task of text classification using graph-based models. This problem, although it is highly relevant to the formulation studied here, it is not discussed at all in the paper. In the past, several approaches have been introduced for graph-based text classification —  none of them is mentioned in the paper. For instance, one of the very first papers for this task “Graph Convolutional Networks for Text Classification” at AAAI ’19, follows a similar formulation where a graph composed of document nodes is constructed. Is there any specific reason why such methodologies have not been used?

* Despite targeting scalability, the paper does not discuss the convergence of the proposed model. The training step is quite complex, therefore studying the convergence of the model is important.

* The settings of structure-free inductive learning are not very clearly presented. Since GLEM assumes that there is a graph structure that captures the relationships among text nodes, it is not straightforward to extend it to an inductive setting. How is this problem tackled here?


**Summary Of The Paper:**

The paper proposes a methodology to perform classification on text-attributed graphs. This is an important problem that arises in many practical applications where the nodes of the graph are associated with some textual information. The paper studies a methodology that combines pre-trained language models (LM) and GNNs. The proposed framework, called GLEM, is trained using a variational EM algorithm. Specifically, fixing the GNN model which characterizes the global label distribution, the parameters of the LM are updated in the E-step. Then, in the M-step, the parameters of the GNN are optimized. The paper puts emphasis on the choice of the likelihood function to deal with scalability constraints. The proposed methodology is evaluated on several node classification tasks.

**Summary Of The Review:**

Overall, I believe it’s an interesting formulation of the node classification task on text-attributed nodes. Nevertheless, I still have some concerns about the absence of baseline models. This point needs further clarification from the authors.

---

> ### Author Response · Authors · 2022-11-12
> **Response to Reviewer 5qjT**
>
> We sincerely appreciate your kind comments and your insightful suggestions.
> Regarding the weaknesses, we hope our point-to-point response can address your concerns.
>
> **W1. Regarding previous studies on graph-based text classification**
>
> Thank you for pointing out these papers! Indeed, the paper also proposes a GNN method for text classification, yet there are some key differences between this paper and ours. Specifically, this paper studies the standard text classification problem, where there are no edges between the texts to be classified. By contrast, we focus on text-attributed graphs, where some edges between texts are provided and our goal is to use both textual and structural information for classification.
> We have incorporated the discussion in the updated draft. See the blue text in the related work section.
>
> **W2. Framework complexity and convergence of the proposed model**
>
> Complexity and convergence are good points!
> As derived in Section 4, the GLEM framework finally reached a simple form, where LMs and GNNs are trained separately by optimizing a pseudo-label-enhanced loss in Eq.8 and Eq.9. Since only a pseudo-label-based loss term is added compared with the original cross-entropy loss, we kindly argue that GLEM is a relatively simple yet effective framework without adding much complexity to the training of both modules.
> Regarding convergence, GLEM converges within a few (sometimes even one) iters. Please kindly check Section 5.5 for a more detailed discussion.
>
> **W3. How structure-free inductive experiments are performed**
>
> We are sorry about the confusion. The structure-free inductive experiments are performed in the following way.
> During the training phase, we consider a collection of nodes, and their text attributes and the edges between them are both given. These nodes are used as training data. During the test phase, we also consider a collection of nodes, but only their text attributes are observed, and thus the problem becomes a standard text classification task.
>
> **Other suggestions**
>
> Regarding the suggestions, your understanding of GLEM-GNN and GLEM-LM is correct. We sincerely appreciate these valuable suggestions and have revised the paper.

---

### Decision · Program_Chairs · 2023-01-20

**Decision:**

Accept: notable-top-5%

**Justification For Why Not Higher Score:**

N/A.

**Justification For Why Not Lower Score:**

Among all the papers under me, this is the best paper. The paper also addresses an important issue with a nice solution (involving both PLM and GNN). This paper deserves to be a highlight at the conference.

**Metareview: Summary, Strengths And Weaknesses:**

This paper studies the problem of learning representations for large-scale text-attributed graphs. The authors propose a novel variational EM algorithm incorporating textual and structural information in such graphs.

All reviewers agree this is a nice paper, with sound theoretical and empirical components, integrating the popular pre-trained language models and the graph neural networks. The execution of the work was also excellent, with comprehensive experiments and analyses. The paper’s quality is high.

There are some concerns about the experiments (baselines, settings, ablations, etc.), clarity, and reproducibility. The authors have addressed them well in the rebuttals.

**Note From Pc:**

if the above contains the word "oral" or "spotlight" please see: "oral" presentation means -> notable-top-5% and "spotlight" means -> notable-top-25%. As stated in our emails, we are disassociating presentation type from AC recommendations